# Production of a β-Glucosidase-Rich Cocktail from *Talaromyces amestolkiae* Using Raw Glycerol: Its Role for Lignocellulose Waste Valorization

**DOI:** 10.3390/jof7050363

**Published:** 2021-05-06

**Authors:** Juan A. Méndez-Líter, Laura I. de Eugenio, Neumara L. S. Hakalin, Alicia Prieto, María Jesús Martínez

**Affiliations:** Centro de Investigaciones Biológicas Margarita Salas (CIB), Consejo Superior de Investigaciones Científicas (CSIC), Ramiro de Maeztu 9, 28040 Madrid, Spain; jmendez@cib.csic.es (J.A.M.-L.); lidem@cib.csic.es (L.I.d.E.); nhakalin@gmail.com (N.L.S.H.)

**Keywords:** biodiesel by-products, cellulases, fungi, hydrolases, wastes

## Abstract

As β-glucosidases represent the major bottleneck for the industrial degradation of plant biomass, great efforts are being devoted to discover both novel and robust versions of these enzymes, as well as to develop efficient and inexpensive ways to produce them. In this work, raw glycerol from chemical production of biodiesel was tested as carbon source for the fungus *Talaromyces amestolkiae* with the aim of producing enzyme β-glucosidase-enriched cocktails. Approximately 11 U/mL β-glucosidase was detected in these cultures, constituting the major cellulolytic activity. Proteomic analysis showed BGL-3 as the most abundant protein and the main β-glucosidase. This crude enzyme was successfully used to supplement a basal commercial cellulolytic cocktail (Celluclast 1.5 L) for saccharification of pretreated wheat straw, corroborating that even hardly exploitable industrial wastes, such as glycerol, can be used as secondary raw materials to produce valuable enzymatic preparations in a framework of the circular economy.

## 1. Introduction

The current generation of wastes at worldwide level is a direct consequence of the inefficient use of natural resources in human activities. Industry has traditionally focused on the production of goods and services, consuming raw materials and generating wastes that remain in the environment. Valorization of the wide variety of mankind-produced waste has become an essential task for a sustainable world [1]. On the other hand, the need for energy and commodities is continuously increasing, while fossil fuels reserves are declining and finite. Therefore, it is imperative to learn how to use waste from other processes as secondary raw materials to produce alternative energy sources and materials.

Many carbon-rich industrial or urban residual streams are susceptible to re-valorization into value added products. In this sense, biodiesel is a renewable biofuel that can be produced from vegetal oils, animal fat, synthetic algae, or cooking oil waste [2]. In fact, fuels from renewable biomass are the cleanest sources of energy, having the potential to reduce the amount of CO_2_, nitrogen oxides (NOx), sulfur oxides (SOx), CO, and other hazardous particles released to the atmosphere compared with fossil diesel [3]. One of the main problems derived from industrial production of biodiesel is the generation of huge amounts of glycerol as byproduct (10:1 ratio). The global biodiesel production has increased during the last decade, reaching more than 30.8 Mm^3^ in 2016, and its production will achieve 41 Mm^3^ in 2022 [2], which will be translated into millions of m^3^ of glycerol. Then, in the global market, glycerol has become an abundant raw material to be used, mainly in the chemical industry. However, the current industrial process for synthesis of biodiesel generates a by-product containing about 70–80% glycerol together with water, catalysts, fatty acids, and salts. This makes the commercialization and valorization of this by-product difficult, since the glycerol required as raw material for pharmaceutical, cosmetic, and food industries requires high purity standards.

As the structural component of many lipids, glycerol is widely found in nature and some microorganisms can naturally use it to grow in the absence of additional carbon sources. These organisms can be potent tools for bioconversion of glycerol from biodiesel manufacturing into valuable microbial products, such as enzymes with industrial interest, medicinal drugs, antibiotics, and other chemicals [4,5,6]. Glycerol has substituted typical carbohydrates, such as sucrose or glucose, in industrial fermentations for production of 1,3-propanediol, dihydroxyacetone, succinic acid, propionic acid, citric acid, pigments, polyhydroxyalkanoates, biosurfactants, docosahexaenoic acid, lipids, or syngas [7].

On the other hand, lignocellulosic biomass can be converted into fermentable sugars by the synergistic action of a battery of enzymes including, exo- and endoglucanases and β-glucosidases, as well as other auxiliary enzymes [8]. Among them, β-glucosidases are essential to release free glucose, but commercial cellulase preparations, especially those from *Trichoderma* spp., are usually deficient in this activity and must be supplemented to increase their efficiency in saccharification processes. In this context, *Talaromyces amestolkiae* has been studied for its ability to secrete a powerful cellulosic cocktail with several β-glucosidases depending on the carbon source in the culture media [9]. Analysis of the secretomes released in those media revealed two mayor β-glucosidases. One of them, named as BGL-2, was induced exclusively by cellulosic substrates [10], while the other, BGL-3, was produced independently of culture carbon source [11]. BGL-3 is a versatile 1,4-β-glucosidase that has proved to be very efficient for saccharification of wheat straw, but also highly active over 1,3-β-glucose bonds [11].

In this work we propose the use of raw glycerol as an appropriate carbon source to produce efficient enzyme cocktails with high β-glucosidase levels using the fungus *T. amestolkiae*. The efficiency of these cocktails for degradation of lignocellulosic residues was evaluated, and plant biomass valorization is discussed in the framework of a circular economy approach.

## 2. Materials and Methods

### 2.1. Chemicals, Microorganisms, Culture Media, and Crude Enzyme Extract Production

Raw glycerol was obtained by separation of biodiesel after chemical transesterification of waste cooking oil and sodium metoxide [12]. It was directly supplied as carbon source in a basal medium for fungal growth. All other chemicals used in this work were of reagent grade and purchased from Sigma-Aldrich.

*T. amestolkiae* was isolated from cereal waste and included in the collection of the Institute Jaime Ferrán of Microbiology (IJFM) with the number A795. For spore production, the fungus was grown in potato dextrose agar (PDA, Difco) for 5 days at 28 °C, and a suspension was obtained by placing 1 cm^2^ agar in 5 mL of a 1% NaCl solution with 0.1% of Tween 80. Pre-inocula were prepared by inoculating 200 µL of this homogenous spore suspension in 250 mL flasks containing 50 mL of corn steep solid (CSS) medium (40 g/L glucose, 0.4 g/L FeSO_4_·7H_2_O, 9 g/L (NH_4_)_2_SO_4_, 4 g/L K_2_HPO_4_, 26.3 g/L corn steep solid, 7 g/L CaCO_3_, and 2.8 mL/L soybean oil), previously autoclaved at 110 °C for 20 min. The cultures were incubated at 28 °C and 250 rpm for 5 days.

Enzyme production was carried out in 1 L flasks containing 200 mL of Mandels minimal medium (2.0 g/L KH_2_PO_4_, 1.3 g/L (NH_4_)_2_SO_4_, 0.3 g/L urea, 0.3 g/L MgSO_4_·7H_2_O, 0.3 g/L CaCl_2_, 5 mg/L FeSO_4_·7H_2_O, 1.6 mg/L MnSO_4_·H2O, 1.4 mg/L ZnSO_4_·7H_2_O, and 1 g/L Bacto peptone) [13], supplemented with 0.5%, 1%, and 2% (*w/v*) raw glycerol as the carbon source, both unbuffered (pH 4.5) and buffered with 100 mM sodium phosphate pH 6. After autoclaving at 121 °C for 20 min, which was also useful for glycerol sterilization, the flasks were inoculated with 2 mL of the pre-inoculum, and incubated at 28 °C and 250 rpm for 10 days, taking 1.5 mL aliquots daily for analytical determinations. After that time, cultures were filtered, concentrated, and dialyzed against a pH 4 sodium acetate buffer at 10 mM to produce the enzymatic cocktails. Concentration was achieved using an ultrafiltration cell (Amicon, Merck–Millipore, Kenilworth, NJ, USA) with a polysulfone membrane (3 kDa cut-off, Millipore). The process was carried out with gentle stirring, using nitrogen gas for pressure (2 bar).

### 2.2. Determination of Biomass, Proteins, Enzyme Activity, and Glycerol Consumption

Aliquots from the cultures were centrifuged and filtered. The content of *T. amestolkiae* biomass in the different samples was determined from the dry weight of the lyophilized pellets. The other parameters were measured in the supernatants. Protein concentration was evaluated by using the commercial Bio-Rad Bradford assay (dye reagent concentrate) (Bio-Rad, CA, USA) and bovine serum albumin as the standard. β-glucosidase activity was tested following the hydrolysis of 0.1% (*w/v*) *p*-nitrophenyl-β-D-glucopyranoside (*p*NPG, Sigma, St. Louis, MO, USA) at 50 °C, in a 100 mM sodium acetate buffer at pH 4. The reaction was stirred at 1200 rpm for 10 min in a Thermo shaker (T-100, SC-24 block), stopped by adding 2% (*w/v*) (1.42% final concentration) of Na_2_CO_3_, and the *p-*nitrophenol (*p*NP) released was spectrophotometrically measured at 410 nm (ε_410_ = 15,200 M^−1^ cm^−1^). One unit of β-glucosidase activity was defined as the release of 1 μmol of *p*NP per minute. Avicelase (as indicative of exocellulase activity) and β-1,4-endoglucanase activities were measured by determining the release of reducing sugars by the Somogyi–Nelson method, as previously described [9], with 1% Avicel (Merck–Millipore) or 3% low viscosity carboxymethylcellulose (Sigma), respectively, as substrates. One unit of activity was defined as the corresponding to the release of 1 μmol of reducing sugar per minute. To determine glycerol consumption, the Glycerol Assay Kit (Sigma) was used according to the manufacturer’s instructions.

All assays were performed in triplicate, and significant differences between samples were explored using Student’s *t*-test, considering a *p* value < 0.05 as the limit for considering differences.

### 2.3. Proteomic Analysis of T. amestolkiae Secretomes

For proteomic analysis, samples from culture supernatants of *T. amestolkiae* grown in Mandels medium plus 1% raw glycerol were analyzed. The process was carried out as previously described [9], with some minor variations. In brief, 2 mL of the liquid supernatants from 7-day-old cultures were centrifuged at 20,000× *g* for 10 min to remove mycelium. Then, aliquots containing 5 μg of proteins were first dissolved in SDS-PAGE loading buffer, denatured, loaded in a 12% SDS gel, and run for 10 min, in order to remove non-protein compounds before the proteomic analysis. The whole protein band was excised from the gel, cut in small pieces, and digested with trypsin overnight at 37 °C. Then the peptides mixture was extracted and analyzed by nano-HPLC–MS/MS.

A NanoEasy HPLC (Proxeon Biosystems, Odense, Denmark), coupled to a nano-electrospray ion source (Proxeon Biosystems), was used for all peptide separations. Samples were loaded onto a C18-A1 ASY-Column 2 cm precolumn (Thermo Scientific, MA, USA) and then eluted onto a Biosphere C18 column (C18, inner diameter 75 µm, 15 cm long, 3 µm particle size, Nano Separations, Nieuwkoop, Netherlands) as previously reported. Full-scan MS spectra (*m/z* 300–1800) were acquired in the LTQ-Orbitrap Velos in the positive ion mode (Appendix A).

Mass spectra files were searched in a specific database against the *T. amestolkiae* genome. Precursor and fragments mass tolerance were set to 10 ppm and 0.5 Da, respectively. Search parameters included a maximum of two missed cleavages allowed, carbamidomethylation of cysteines as a fixed modification, and oxidation of methionine as a variable modification. Peptides were validated through the algorithm Percolator [14], and only those with high and medium confidence were admitted (FDR 0.05). Protein identifications were only accepted if they contained at least two identified peptides. Results were inferred from data obtained from two technical replicates from two different biological samples. Relative quantification of the most abundant proteins in the samples analyzed was calculated from the number of peptide spectrum matches (PSMs) corresponding to each protein [15,16].

### 2.4. Saccharification of Pretreated Wheat Straw

The efficiency of *T. amestolkiae* enzyme cocktails as β-glucosidase supplements for saccharification was studied. For this, wheat straw subjected to three different pretreatments was used (Figure 1A): (i) acidic wheat straw (abbreviated as AcSE), obtained by steam explosion in the presence of dilute sulfuric acid (kindly provided by Abengoa Bioenergía, Babilafuente, Salamanca); (ii) neutral wheat straw (abbreviated as SE), obtained by steam explosion in the presence of water, also provided by Abengoa Bioenergía; (iii) and alkaline wheat straw (abbreviated as AP), prepared as follows: 5 g of wheat straw was incubated with 100 mL of 2% NaOH for 40 min and 110 °C using an autoclave, the solid material was then washed three times with water, the pH was adjusted to 5 with sulfuric acid, lyophilized, and stored. The composition of AP wheat straw was analyzed by sequential fractionation. In this process, the contents of cellulose, hemicellulose, and Klason lignin were determined through a quantitative acid hydrolysis [17]. In brief, the solid residue (300 mg) was treated for 1 h with 3 mL of 13.5 M sulfuric acid (H_2_SO_4_) at 30 °C and 150 rpm. Then, H_2_SO_4_ was diluted to 0.5 M and heated at 110 °C for 1 h. Sugars in the liquid fraction were identified and quantified by gas chromatography [18]. The slurry remaining after acid hydrolysis (Klason lignin) was washed until neutrality, oven dried at 55 °C, and weighted. All analyses were carried out in triplicate.

For saccharification, 100 mg of AcSE, SE, or AP wheat straw was incubated in 2 mL 100 mM sodium acetate buffer at pH 4 for 96 h, at 40 °C and 1200 rpm in the presence of different enzymatic combinations (Figure 1B):-2 U of β-glucosidase activity from Celluclast 1.5 L (Novozymes, Copenhagen, Denmark). This is a commercial cocktail, rich in cellobiohydrolase and endoglucanase activities, and was used for comparison purposes;-1 U of β-glucosidase activity from Celluclast 1.5 L + 1 U of β-glucosidase activity from *T. amestolkiae* enzymatic crudes obtained in this work, using 1% glycerol as the carbon source.

## 3. Results

### 3.1. BGL Production and Growth of T. amestolkiae in Media with Raw Glycerol

The ascomycete *T. amestolkiae* has been proven as an excellent producer of robust and efficient cellulolytic enzymes in media with different carbon sources [9]. This finding suggested the convenience of testing cheaper carbon sources to obtain these cocktails, since they are rich in β-glucosidases, the key enzymes for cellulose saccharification.

In this work, we studied the use of different concentrations of raw glycerol, an abundant waste generated from biodiesel production, to produce added-value enzyme cocktails. To analyze the potential of this carbon source, we followed the evolution of fungal growth and the extracellular levels of β-glucosidase over 10-day-old cultures (240 h). As observed in Figure 2A, *T. amestolkiae* cultures in unbuffered media reached maximal biomass values around 24 h, regardless of the glycerol concentration assayed, although total biomass was higher as the glycerol concentration increased. The highest β-glucosidase levels were detected in cultures containing 1% or 2% glycerol (Figure 2B). BGL levels were low in the first 24–48 h, and then they started to rise uninterruptedly at a high rate between 72 h and 168 h. Surprisingly, both cultures produced around 8 U/mL at the final incubation time, despite that the fungal biomass was higher with 2% glycerol. Searching for an explanation to this result, it was observed that the pH was more acidic in the culture with 2% glycerol (pH 5) than in the other cultures (pH 6–6.5). The acidification was detected at 168 h and maintained until the end of the incubation period.

Thus, to investigate the effect of pH on BGL production, we repeated the experiment buffering all media with phosphate pH 6 to maintain the pH as constant across the incubation period. In these conditions, BGL activity and fungal growth went in parallel, and the highest values for both were detected in cultures with 2% glycerol (Figure 3A). It should also be noticed that the total amount of secreted proteins detected in 2% glycerol unbuffered cultures was considerably lower than in the buffered ones (0.10 and 0.17 mg/mL, respectively). These results suggest that pH should be strictly controlled in the cultures, since its value is related to the levels of secreted proteins. Buffered and unbuffered cultures with 0.5% and 1% glycerol followed a very similar behavior, where the pH was maintained in a pH range of 6–6.5.

On the other hand, these results confirm that *T. amestolkiae* does not require any specific inducer (cellulosic or non-cellulosic) to produce high amounts of β-glucosidases, as previously reported. This is probably due to the presence of BGL-3, an efficient β-glucosidase secreted under carbon starvation conditions [11]. BGL-3 is a β-1,4 glucosidase with a prominent activity on β-1,3 glucans. The high levels of this protein detected during secondary metabolism in the cultures may be due to its activity degrading the β-1,3-glucan released from the fungal cell wall during autolysis. As expected, the amount of other cellulolytic activities like avicelase or β-1,4-endoglucanase was negligible (data not shown), since these enzymes are usually released upon induction by cellulosic substrates.

Only a handful of studies have reported the production of fungal β-glucosidases using glycerol as the carbon source. For *Penicillium echinulatum*, glycerol was the preferred carbon source over cellulose, sugar cane bagasse (pretreated by steam explosion), or glucose [19]. *Aspergillus niger* NRRL 3112 could produce considerable amounts of this activity when grown on wheat bran and glycerol as co-substrates [20]. In other cases, the use of glycerol was not suitable for β-glucosidase production, as reported in *Penicillium funiculosum* [21]. *Trichoderma reesei* has been the most used fungus for producing cellulolytic cocktails, although it is well documented that the amount of β-glucosidase released by this fungus is insufficient for efficient hydrolysis of lignocellulosic biomass [22]. In this sense, one of the main advantages of the use of *Aspergillus* spp., *Penicillium* spp., or *Talaromyces* spp. lies in their higher secretion of β-glucosidase activity [22]. Many published reports allow us to compare its production among these fungal genera, mainly using crystalline cellulose as the carbon source. Different *Aspergillus* sp. were secreted between 0.5 U/mL and 2.5 U/mL of β-glucosidase [23,24]. *Penicillium* species, like *Penicillium brasilianum* or *Penicillium decumbens,* reached a total activity of 3.5 U/mL and 2.39 U/mL, respectively [25,26]. Higher β-glucosidase levels, from 10 to 150 U/mL, have been reported in strains of *P. funiculosum, Penicillium occitanis,* or *Penicillium verriculosum* [21,27,28] using cellulosic inducers. It should be noted that the amounts of BGL observed in *T. amestolkiae* cultures in media with raw glycerol as the sole carbon source are among the highest reported to date. The fact that this high production was achieved using a by-product of the biodiesel industry makes it even more interesting.

Glycerol consumption was monitored and related with the increase of BGL activity (Figure 3B). In cultures with 0.5% glycerol, the depletion of the carbon source occurred in 24 h, while in those with 1% and 2%, it took around 48 h. Low β-glucosidase levels were detected before glycerol exhaustion, but the activity started to rise slowly once consumed, and increased faster when carbon starvation conditions were fully established, expanding over time. The data in Figure 3B indicate that the maximal exploitation of glycerol for BGL production was obtained with 2% glycerol, so we chose this culture as the best model to produce BGL-rich cocktails.

The secretion of β-glucosidases in aged fungal cultures has been scarcely studied, although it would be interesting to know to what extent their production under carbon starvation is common among filamentous fungi. In the case of *T. amestolkiae*, it has been proved that induction of β-glucosidase BGL-3 was detected during carbon shortage. This enzyme is highly active on β-1,3 polysaccharides, suggesting its possible physiological role in cell wall metabolism during carbon starvation, releasing products that could be used as an alternative carbon source [11]. Some authors have reported the production of glycosyl hydrolases after carbon depletion in cultures of *A. niger* [29,30], but their finding was not discussed as a potential way to produce these enzymes. In any case, the versatility of *T. amestolkiae* to produce high levels of BGLs from different carbon sources, including some important industrial residues, should be exploited.

### 3.2. Fungal Secretome Analysis

An extracellular proteomic analysis of 7-day-old cultures was carried out to elucidate the protein profiles of *T. amestolkiae* cultures with 1% raw glycerol. The samples were subjected to tryptic digestion and LC–MS/MS of the whole peptide mixtures produced. The number of proteins identified was 148, similar to that reported in cultures with glucose or cellulosic substrates [9]. In the TIC scan of the proteomic analysis (Appendix A), an archive with al peptides was detected (Appendix A), and a list with the 148 proteins detected (Appendix A) can be found in the Appendix A of the article.

According to KOG, the functional role of most of the proteins identified in cultures with 1% raw glycerol was related to carbohydrate metabolism and transport (55.0%), followed by enzymes involved in amino acid metabolism and transport (Table 1). These data also agree with those previously published for *T. amestolkiae* secretomes with glucose as the carbon source [9]. As the main objective of this work was to produce BGLs from contaminated glycerol waste, we started our analysis classifying the CAZymes (carbohydrate active enzymes), finding 80 out of 141 detected proteins (54%). Among them, the most abundant were glycosyl hydrolases (GHs) from GH3, GH15, GH31, and GH55 families (Table 2). It is worth noting that GH15 enzymes were predominant in cultures with glucose as the carbon source [9], while in media with glycerol, GH3 proteins were the most abundant. The family GH3 is normally associated with enzymes degrading polysaccharides from lignocellulosic biomass, including many β-glucosidases, which supports the suitability of using this carbon source for β-glucosidase production. As expected, BGL-3 was found in the studied secretome (Table 3).

As already mentioned, this enzyme is secreted under carbon starvation [11], but other GHs potentially related with carbon depletion were also detected. This is the case of glucoamylases (GH15), α-glucosidases (GH31), and exo-β-1,3-glucanases (GH55), related to the fungal autophagy process, since 1,3-β-glucans and 1,4-α-glucan are components of the cell wall of many *Talaromyces* species [31]. Besides, it is noticeable that some of the most abundant proteins are related with the fungal autolysis processes. For example, cathepsin, which was detected in every condition, is a well-known protease that could be used by *T. amestolkiae* for degrading proteins and using its components as nutrients. The amino acids produced by cathepsin can be utilized by glutaminase, which has been related with glutamine metabolism under starvation conditions, when no other carbon source is present in the medium.

Regarding β-glucosidases, the most important result was the confirmation of the presence of BGL-3 as the most abundant extracellular protein. This is in good agreement with our previous results describing its production upon carbon exhaustion [9]. Besides, one GH1, recently characterized [32], and four GH3 β-glucosidases were also identified (Table 4). This profusion of β-glucosidases could contribute to explain why this activity was so high in this medium, as compared with other conditions

In summary, the proteomic analysis of *T. amestolkiae* growing in raw glycerol confirmed the presence of BGL-3 and revealed that other interesting enzymes, including different β-glucosidases, can also be obtained for different biotechnological applications by exploiting the carbon starvation metabolism of *T. amestolkiae*.

### 3.3. Wheat Straw Saccharification

To check the efficiency of the novel enzyme cocktail obtained in this work for saccharification of lignocellulosic biomass, we used it as supplement of β-glucosidase activity for Celluclast 1.5 L (a basal cellulolytic preparation, deficient in this activity). Since lignocellulosic substrates have different characteristics, the most effective pretreatment and enzyme cocktail should be used, considering the properties of each raw material. Wheat straw is one of the most important agricultural residues worldwide. According to Iskalieva et al. [33], it is mainly composed of cellulose (38.8%), hemicelluloses (39.5%), and lignin (17.1%). In the present work, three different pretreatments were used to make wheat straw cellulose more accessible: steam explosion (SE), steam explosion in the presence of dilute sulfuric acid (AcSE), and alkaline pretreatment (AP) (Figure 1). Steam explosion is one of the most commonly used methods for lignocellulose pretreatment, deconstructing and modifying part of the lignin, and solubilizing hemicellulose [34,35]. However, although cellulose is more accessible in the steam-exploded biomass, compounds derived from the partial hydrolysis of sugars and lignin remain embedded, producing adverse effects on downstream processes, including enzyme inhibition [36]. The use of dilute sulfuric acid could theoretically avoid this problem, since it helps to remove a bigger part of the lignin, but some of the polymer is always present in the material. On the other hand, the alkaline pretreatment with NaOH has been reported to increase cellulose digestibility by inducing a big reduction of the lignin content [37]. These pretreatments affect the composition of the material used for saccharification differently (Table 5).

In the case of alkaline pretreatment, most lignin was solubilized and removed, leaving a solid fraction enriched in polysaccharides. The two steam explosion treatments gave wheat straw slurries with different cellulose and hemicellulose contents. The AcSE treatment (dilute sulfuric acid) more strongly affected hemicellulose and lignin, forming compounds that produce adverse effects on downstream processes [38]. The results of the saccharification of pretreated wheat straw can be seen in Figure 4.

The combination of Celluclast 1.5 L with the *T. amestolkiae* cocktail improved the saccharification yield of steam-exploded wheat straw (with or without sulfuric acid), compared with Celluclast 1.5 L alone, reaching a cellulose conversion of 65% for SE and of 86% for AcSE. The commercial Celluclast 1.5 L contains an array of cellulolytic and hemicellulolytic activities in addition to BGL, which could theoretically improve saccharification. However, cellulose degradation with this cocktail was 41% for SE and 53% for AcSE.

On the other hand, the saccharification of alkali-pretreated wheat straw was similar with both cocktails (86% for only Celluclast and 89% when supplemented with β-glucosidase from *T. amestolkiae*). One possible reason for this is the virtual lack of lignin in the alkali-pretreated wheat straw. It has been reported that lignin could trigger an irreversible cellulase inactivation, even when using pretreated lignocellulosic biomass, by two possible mechanisms: (i) forming a physical barrier that prevents enzyme access, and (ii) by non-productively binding of cellulolytic enzymes [34,39]. The non-productive binding of enzymes to lignin is recognized as a problem to overcome for improving saccharification efficiency of pretreated lignocellulosic biomass to fermentable sugars. A feasible explanation for our results is that the enzymes from the BGL-rich cocktails from *T. amestolkiae* were better than those contained in Celluclast 1.5 L, avoiding the non-productive binding to lignin. This observation perfectly correlates with some studies that suggest that β-glucosidases and xylanases could avoid non-productive binding to lignin better than other cellulolytic enzymes [40]. Besides, it is remarkable that BGL-3, the most abundant cellulase in *T. amestolkiae* cocktails, possesses a fibronectin III domain (FnIII) [11]. Lima et al. [41], postulated that the FnIII domain strongly interacts with lignin fragments, preventing unproductive binding of cellulases to the lignin. Taking this into consideration, BGL-3 rich cocktails may be advantageous for the saccharification of pretreated residues where lignin is present, representing an effective and cheap alternative to current commercial enzymatic preparations.

## 4. Conclusions

In this work, we demonstrated the potential of the fungus *T. amestolkiae* for producing high levels of β-glucosidase activity using raw glycerol as the carbon source. Proteomic analysis of the crudes revealed that, although other minor glycosyl hydrolases were also found in the secretomes, BGL-3 was the main enzyme responsible for this behavior. This novel enzymatic cocktail was efficiently used to supplement the commercial cellulolytic preparation Celluclast 1.5 L, increasing the glucose yield from wheat straw pretreated by steam explosion. This work set the foundations for expanding the industrial applications of *T. amestolkiae* and exploiting its efficient metabolism, able to use inexpensive carbon sources and to produce β-glucosidases and other value-added enzymes under carbon starvation.

## Figures and Tables

**Figure 1 jof-07-00363-f001:**
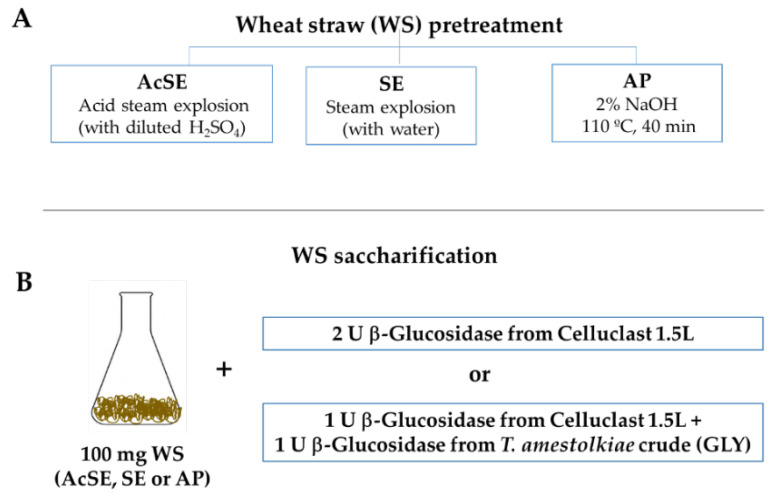
(**A**) Scheme of different wheat straw pretreatments used for enzymatic saccharification and (**B**) enzymatic doses used in the saccharification experiments.

**Figure 2 jof-07-00363-f002:**
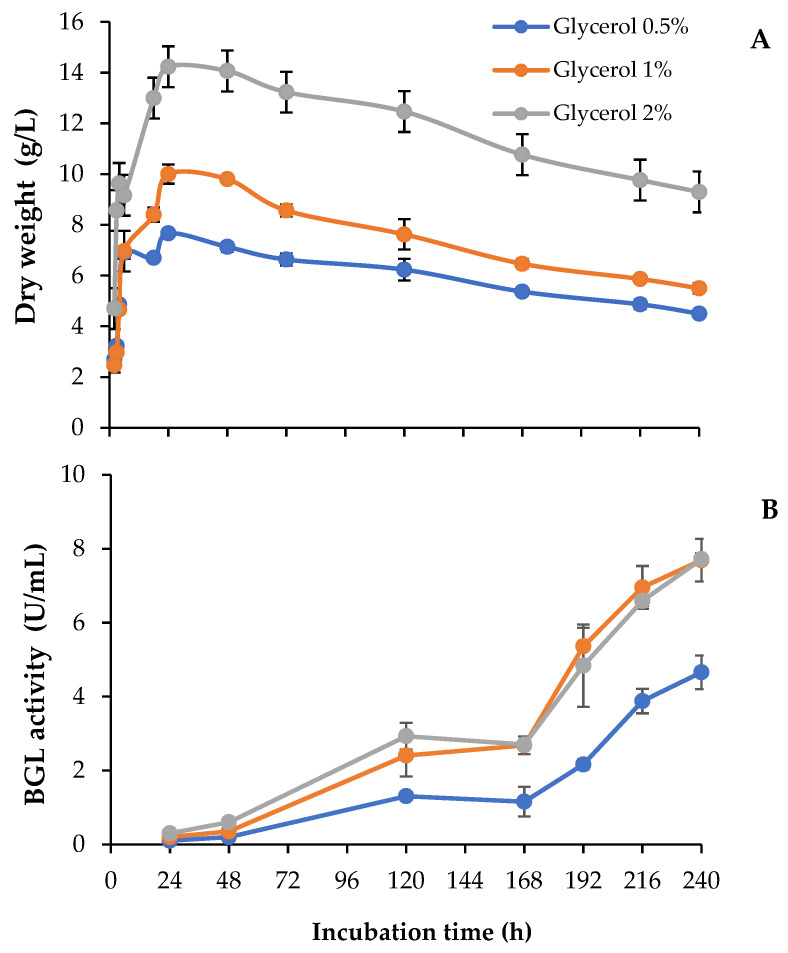
(**A**) Biomass and (**B**) β-glucosidase activity of *T. amestolkiae* growing in unbuffered cultures with raw glycerol (0.5%, 1%, and 2%), as the carbon source. All assays were performed in triplicate.

**Figure 3 jof-07-00363-f003:**
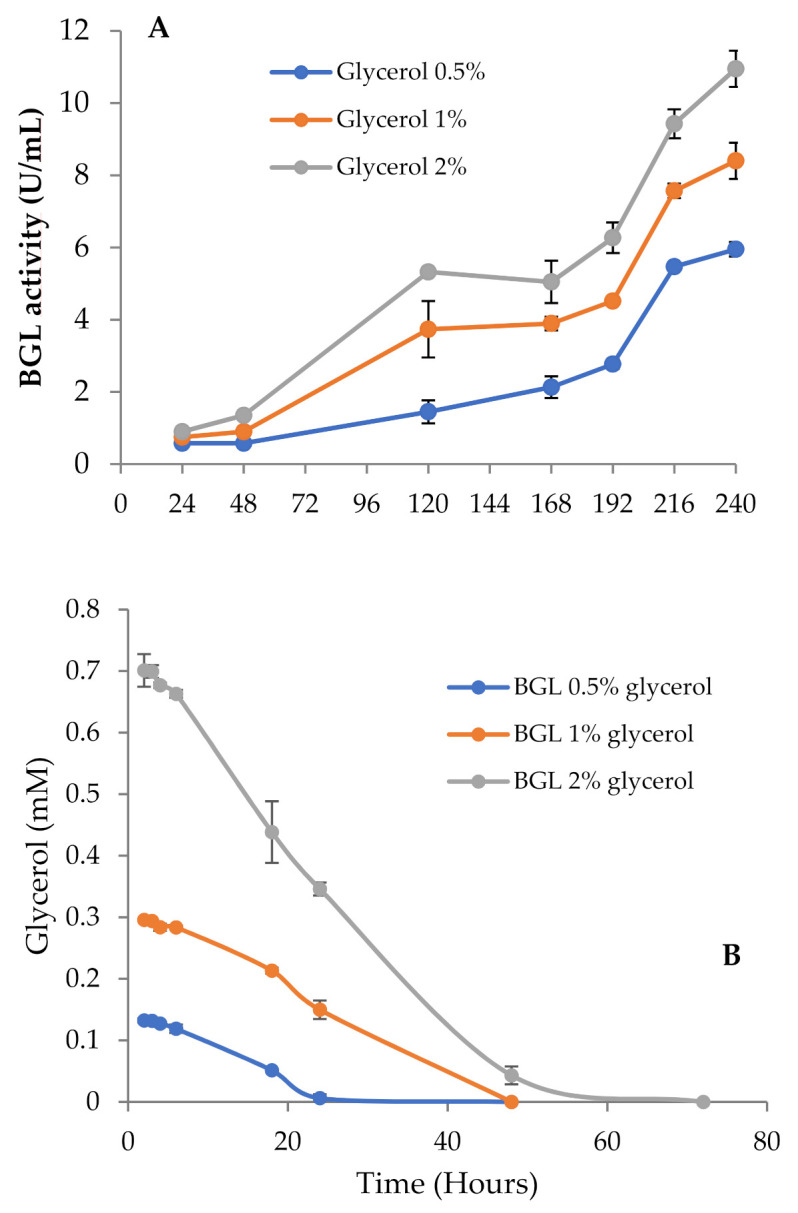
(**A**) β-glucosidase activity of *T. amestolkiae* growing in buffered cultures with raw glycerol (0.5%, 1%, and 2%), as the carbon source. (**B**) Glycerol consumption by *T. amestolkiae* in the buffered cultures. All assays were performed in triplicate.

**Figure 4 jof-07-00363-f004:**
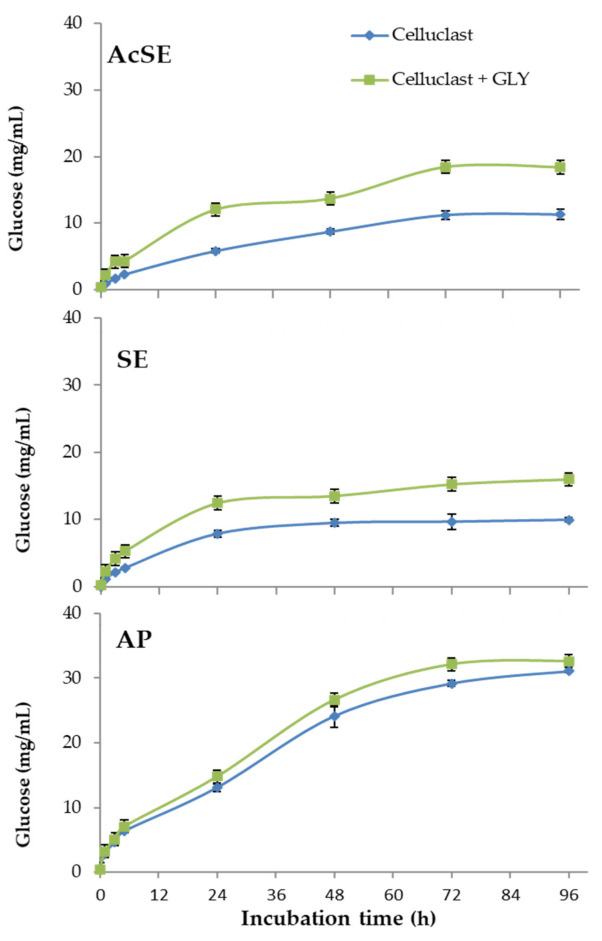
Saccharification of pretreated wheat straw (AcSE: acidic steam explosion; SE: steam explosion; AP: alkali pretreatment) with Celluclast 1.5 L (basal cocktail) and Celluclast 1.5 L supplemented with the *T. amestolkiae* cocktail obtained with raw glycerol as the carbon source (GLY). All assays were performed in triplicate.

**Table 1 jof-07-00363-t001:** Functional classification of the proteins identified in the secretome of 7-day-old *T. amestolkiae* cultures with raw glycerol as the carbon source, compared to those obtained in the same medium with glucose [9].

	% PSM	
	Glycerol	Glucose
A—RNA processing and modification	0.63	0.22
C—Energy production and conversion	2.72	4.90
E—Amino acid metabolism and transport	13.11	10.38
F—Nucleotide metabolism and transport	0.74	1.42
G—Carbohydrate metabolism and transport	55.01	65.16
I—Lipid metabolism	0.09	0.17
M—Cell wall/membrane/envelop biogenesis	0.77	3.61
O—Post-translational modification. protein turnover. chaperone functions	1.96	1.71
Q—Secondary structure	2.55	0.91
R—General functional prediction only	4.81	1.89
S—Function unknown	5.79	3.71
T—Signal transduction	4.13	4.86

**Table 2 jof-07-00363-t002:** Glycosyl hydrolase families identified in secretomes from 7-day-old *T. amestolkiae* cultures growing in raw glycerol, compared to those obtained in the same medium with glucose as the carbon source [9].

	% PSM	
GH Family	Glycerol	Glucose
GH2	3.2	1.7
GH3	16.6	16.3
GH13	3.0	4.3
GH15	10.1	28.4
GH18	2.8	0.6
GH20	3.7	4.8
GH27	2.1	1.7
GH31	8.6	11.8
GH35	2.7	1.8
GH55	8.1	3.9
GH71	4.0	0.1
GH72	3.2	1.3
GH92	5.3	1.1
GH127	3.6	3.2

**Table 3 jof-07-00363-t003:** Most abundant extracellular proteins identified in the *T. amestolkiae* secretome obtained from 7-day-old cultures growing in raw glycerol. BGL-3 is indicated in bold.

Accession ID	% PSM (Average)	Predicted Protein Function	Cazyme Family	*M*_w_ (kDa)
**g377 (BGL-3)**	**7.09**	**beta-glucosidase**	**GH3**	**88.7**
g3995	6.47	Glutaminase	-	76.4
g8295	3.59	alpha-glucosidase	GH31	98.6
g2158	3.28	Glucoamylase	GH15	65.2
g9324	3.18	Exo-beta-1,3-glucanase	GH55	84.3
g2140	2.50	Glucoamylase	GH15	67.7
g5915	2.23	non-reducing end β-L-arabinofuranosidase	GH127	68.8
g4076	2.00	hexosaminidase	GH20	67.9
g216	1.77	neutral/alkaline nonlysosomal ceramidase	-	160.0
g9148	1.58	catalase	-	79.1

**Table 4 jof-07-00363-t004:** Main hypothetical BGLs detected identified in the *T. amestolkiae* secretome obtained from 7-day-old cultures with raw glycerol as carbon source.

Accession ID	% PSM (Average)	Cazyme Family	*M*_w_ (kDa)
g377 (BGL-3)	7.09	GH3	88.7
g9150	1.54	GH3	86.5
g8384	0.85	GH1	68.1
g6857	0.79	GH3	109
g3139	0.30	GH3	93.6
g6753	0.09	GH3	81.8

**Table 5 jof-07-00363-t005:** Main components of wheat straw pretreated by steam explosion in water (SE), steam explosion with dilute sulfuric acid (AcSE), or alkaline pretreatment (AP).

	SE	AcSE	AP
Cellulose	49.0%	43.6%	71.8%
Hemicellulose	15.4%	17.1%	24.1%
Lignin	35.6%	39.3%	4.1%

## Data Availability

*T. amestolkiae* whole genome shotgun project is deposited at DDBJ/ENA/GenBank under the accession MIKG00000000.

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
