# Peer review of "Production of a β-Glucosidase-Rich Cocktail from Talaromyces amestolkiae Using Raw Glycerol: Its Role for Lignocellulose Waste Valorization"

_jof, 2021, doi:10.3390/jof7050363_

Round 1
Reviewer 1 Report
General comments
This study proposes the valorization of a residual stream from biodiesel production (raw glycerol) into enzymes that can be used in the production of biofuels. Thus, the application of the biorefinery concept broadens the range of circular economy, providing a very interesting and novel perspective. The experimental design of the work, as well as the manuscript, has been carefully structured, following a straightforward and rigorous way. Before being considered for publication, the authors should address the following minor points.
Specific comments
1) In M&M, please include the composition of the Mandels minimal medium. Is the carbon source of this culture medium the main contributor to the enzyme production cost?
2) In Pag.4 (first paragraph), modify the following sentence: "the solid residue was treated for 1 h with 100 gL−1 72% sulfuric acid (H2SO4)". It may be simpler and more clear to express the composition in mol/L.
3) Please modify Figure 2. Is the time scale on the upper and lower part of the graph the same? In the lower graph, the scale of the x-axis is not numerical, as seen by the difference between the first 24 h, and the space from 24 to 48 h. I would suggest separating the symbols and line graphs, from the bar chart.
4) Include the standard deviation (SD) of the results presented in Figure 3. Are the differences between the hydrolysis with Celluclast 1.5L alone and that supplemented with the T. amestolkiae cocktail significant when the SD is taken into account?
5) In Pag.4-L.180, it is commented that "... both cultures produced around 8 U/mL ...", however, in Figure 1B, the activity is shown in 'mU/mL'. Which one is the correct activity unit? Please, also check the activity units in Figure 2.
6) It would be interesting to carry out the secretome analysis of T. amestolkiae at the end of the fermentation, taking into account that significant production of beta-glucosidase was observed from day 7 to the end of the culture (168 to 240 h).
Author Response
Reviewer 1
General comments
This study proposes the valorization of a residual stream from biodiesel production (raw glycerol) into enzymes that can be used in the production of biofuels. Thus, the application of the biorefinery concept broadens the range of circular economy, providing a very interesting and novel perspective. The experimental design of the work, as well as the manuscript, has been carefully structured, following a straightforward and rigorous way. Before being considered for publication, the authors should address the following minor points.
We appreciate the good consideration of reviewer 1 about our work. Specific corrections are below:
Specific comments
1) In M&M, please include the composition of the Mandels minimal medium. Is the carbon source of this culture medium the main contributor to the enzyme production cost?
Composition of Mandels medium has been added (lines 89-91). Mandels medium is composed by salts and nitrogen sources that have a residual cost compared to the use of carbon sources of limited availability, such as oligosaccharides or microcrystalline cellulose. Our experience working with Mandels medium has been successful, even when using residues for enzymatic induction, as performed in the current work.
2) In Pag.4 (first paragraph), modify the following sentence: "the solid residue was treated for 1 h with 100 gL−1 72% sulfuric acid (H2SO4)". It may be simpler and more clear to express the composition in mol/L.
A sentence has been added (line 162): “In brief, the solid residue (300 mg) was treated for 1 h with 3 mL of 18.4 M sulfuric acid (H2SO4) at 30 °C and 150 rpm, then H2SO4 was diluted to 0.5 M and heated at 110 °C for 1 h.”
3) Please modify Figure 2. Is the time scale on the upper and lower part of the graph the same? In the lower graph, the scale of the x-axis is not numerical, as seen by the difference between the first 24 h, and the space from 24 to 48 h. I would suggest separating the symbols and line graphs, from the bar chart.
Reviewer 1 is completely right. The x axis was not numerical, which could generate misunderstanding of the data. Figure 2 (now figure 3, according to suggestion of rewiever 3) has been modified, eliminating the bars graph, since they represented the same data contained in the A part of the figure. In any case, we think that now the figure is clear enough to comprehend the data.
4) Include the standard deviation (SD) of the results presented in Figure 3. Are the differences between the hydrolysis with Celluclast 1.5L alone and that supplemented with the T. amestolkiae cocktail significant when the SD is taken into account?
We apologize for forgetting to include standard deviation data. The figure 3 (in the revised version Fig 4), has been corrected. The differences found are statically significant, even with the SDs. Student´s-t test was performed for all the assays included in the work.
5) In Pag.4-L.180, it is commented that "... both cultures produced around 8 U/mL ...", however, in Figure 1B, the activity is shown in 'mU/mL'. Which one is the correct activity unit? Please, also check the activity units in Figure 2.
We wanted to present activity data in U/mL units, the use of mU/mL was a mistake. Both figures are now with the same units. We really apologize for this.
6) It would be interesting to carry out the secretome analysis of T. amestolkiae at the end of the fermentation, taking into account that significant production of beta-glucosidase was observed from day 7 to the end of the culture (168 to 240 h).
We are grateful for reviewer observation. The initial analysis was set to day 7 since we wanted to compared our data with T. amestolkiae proteomics information that has been previously published (de Eugenio et al., 2017). The obtaining of high enzymatic loads in short incubation times is really desirable at industrial level for large scale production, since the possibility of contamination is diminished. However, as reviewer suggests, it could be very interesting to study secretomes produced in longer times and we will consider it in the near future.
de Eugenio L, Méndez-Líter J, Nieto-Domínguez M, Alonso L, Gil-Muñoz, J., Barriuso, J., et al. Differential β-glucosidase expression as a function of carbon source availability in Talaromyces amestolkiae: A genomic and proteomic approach. Biotechnol Biofuels. 2017;10:1–14.
Reviewer 2 Report
Authors need to include the following comments:
Full-scan MS spectra (m/z 300-1800) were acquired in the LTQ-Orbitrap in the positive ion mode. Authors need to add the TIC scan for the analysis.
Mass spectra files were searched in a specific database against the T. amestolkiae genome. Which database was used for the searching?
Did the author try higher concentration of glycerol viz 4 or 5%?
“The samples were subjected to tryptic digestion and LC–MS/MS of the whole peptide mixtures produced. The number of proteins identified was 148, similar to that reported in cultures with glucose or cellulosic substrates.” Authors need to included the LC-MS/MS data for the analyzed protein in order to understand the reader because these data will help better understanding of the research.
Author Response
Reviewer 2
We would like to thank reviewer 2 for the suggestions to improve our manuscript.
Authors need to include the following comments:
Full-scan MS spectra (m/z 300-1800) were acquired in the LTQ-Orbitrap in the positive ion mode. Authors need to add the TIC scan for the analysis.
TIC scan of the proteomic analysis has been included in the supplementary data of the article.
Mass spectra files were searched in a specific database against the T. amestolkiae genome. Which database was used for the searching?
The mass spectra *.raw files were searched against an in-house specific database built from T. amestolkiae genome data (deposited at DDBJ/ENA/GenBank under the accession number MIKG00000000, 10408 sequences, 5662098 residues) using the SEQUEST search engine through Proteome Discoverer (version 1.4.1.14, Thermo).
Did the author try higher concentration of glycerol viz 4 or 5%?
We have not tried higher than 2 % glycerol, since impurities contained in biodiesel glycerol have been shown to inhibit fungal growth at high concentrations (Athalye et al., 2009). However, suggestion of reviewer 2 is a very good idea that we will consider for future experiments.
Athalye SK Garcia A, Wen, Z. Use of Biodiesel-Derived Crude Glycerol for Producing Eicosapentaenoic Acid (EPA) by the Fungus Pythium irregular. J. Agric. Food Chem. 2009 57 (7), 2739-2744
“The samples were subjected to tryptic digestion and LC–MS/MS of the whole peptide mixtures produced. The number of proteins identified was 148, similar to that reported in cultures with glucose or cellulosic substrates.” Authors need to include the LC-MS/MS data for the analyzed protein in order to understand the reader because these data will help better understanding of the research.
An excel document with detected peptides has been included as supplementary material of the article. Besides, a table with all the proteins detected has been also added to supplementary material. We hope that with this additions, the proteomic part of the article would be better understood.
Reviewer 3 Report
Dear Authors, here my comments:
- Keywords should be ordered in an alphabetical order.
- Line 42 "many millions" sounds too colloquially
- Line 64 growing on different carbon sources
- Lin 67 in all tested conditions
- Subsection 2.1. please give details about glycerol sterilization.
- Line 83 give a company for PDA medium
- Line 89: why 250 rpm? It seems quite high.
- Line 91 please provide a composition of Mandels minimal medium
- Line 95: provide details about concentration
- Line 98: inappropriate use of word evolution/ Maybe generation or proliferation should be better.
- Line 101: please give a reference to Bradford method. Besides, I think that the Bradford method is far too little sensitive to protein measurement in such analysis. I recommend the use of e.g. Pierce BSA Kit Protein from Sigma-Aldrich.
- Line 104: rpm should be given in x g
- Line 121: How long have you been digested the gel with trypsin? duration of digestion should may influence the results
- Line 133-135: why you particularly choose these proteins modifications?
- The subsection 2.4 is highly confusing. I think that this subsection should be replaced by a graph.
- line 163: saccharification with 2 U of BGL activity of only Celluclast was also used for comparison - I don't understand this sentence.
- Line 181: did you use the statistical tests to compare the obtained data? You should compare the dry weight of fungal biomass between the treatments.
- Line 195: on what basis was it found that the differences are insignificant?
- Line 237: what is your hypothesis about the highest production of enzyme under carbon starvation conditions?
- line 249: why did you choose 1% of glycerol for proteomics? why only one condition?
- What about the fact that activity of enzyme is vulnerable to pH changes? Is it a limitation in its use in waste management? Please consider this in the Conclusion section.
Overall comment: you can improve the charts - maybe add some colors or more varied markers.
Author Response
Reviewer 3
Comments and Suggestions for Authors
Dear Authors, here my comments:
- Keywords should be ordered in an alphabetical order.
The order of the keywords has been modified.
- Line 42 "many millions" sounds too colloquially
We have substracted the word “many” in this sentence (line 41).
- Line 64 growing on different carbon sources
The phrase has been rewritten (line 63)
- Lin 67 in all tested conditions
The phrase has been rewritten (line 66).
- Subsection 2.1. please give details about glycerol sterilization.
Glycerol was sterilized by autoclaving it when added to minimal medium. A phrase has been added in the manuscript to improve understanding in line 94.
- Line 83 give a company for PDA medium
The company is Difco. This information was added to the manuscript (line 82).
- Line 89: why 250 rpm? It seems quite high.
At 250 rpm culture aeration is better for protein production, as it has been described for recombinant protein secretion by several yeast strain and, thus, it has been set as the standard agitation conditions in our laboratory
- Line 91 please provide a composition of Mandels minimal medium
The information has been added to the manuscript. (lines 89-91)
- Line 95: provide details about concentration
Process details have been added to the manuscript (lines 98-100).
- Line 98: inappropriate use of word evolution/ Maybe generation or proliferation should be better.
We appreciate the suggestion of reviewer 3. We have changed this word for “content” (line 102).
- Line 101: please give a reference to Bradford method. Besides, I think that the Bradford method is far too little sensitive to protein measurement in such analysis. I recommend the use of e.g. Pierce BSA Kit Protein from Sigma-Aldrich.
We use the commercial Biorad Bradfor assay. It have been clarify (line 105).
We have also analyzed protein concentration in cultures using the bicinchoninic acid method (Thermo fisher), with similar results. Since Bradford method is easier to perform, we use it routinely in our laboratory, although we always use a second method of protein determination to confirm that the values obtained are correct.
- Line 104: rpm should be given in x g
We haven´t found information about block spin radius for our Thermo shaker. We have added detailed information about thermo shaker and block model (line 109)
- Line 121: How long have you been digested the gel with trypsin? duration of digestion should may influence the results
All the proteomic experiments were performed in collaboration with genomics and proteomics service at Centro de Investigaciones Biológicas Margarita Salas. In the specific case of trypsin digestion, it was performed according to this protocol: Gel pieces were dried, rehydrated with 12.5 ng/µL trypsin in 50 mM ammonium bicarbonate and incubated overnight at 37 °C. A phrase was added in materials and methods section, lines 129-131.
- Line 133-135: why you particularly choose these proteins modifications?
2 misscleavages are chosen to ensure that tryptic peptides are detected even if the trypsin has had some cutting mistakes. As for chemical modifications, carbamidomethylation of cysteines as a fixed modification and oxidation of methionine as a variable modification, are the standard parameters that the genomic and proteomic facilities always use. Since proteins are reduced during the process, free cysteine groups are blocked to prevent reattachment: this is done by carbamidomethylating them by treating the samples with iodoacetamide. It is also very common for methionines to oxidize during proteomic analyses.
- The subsection 2.4 is highly confusing. I think that this subsection should be replaced by a graph.
A graph (figure 1) has been included in order to clarify the saccharification experiment. Also, the text has been rewritten.
- line 163: saccharification with 2 U of BGL activity of only Celluclast was also used for comparison - I don't understand this sentence.
We wanted to compare the ability of our enzymatic preparation with a commercially available cocktail for biomass saccharification. The paragraph has been rewritten for better understanding (lines 171-176).
- Line 181: did you use the statistical tests to compare the obtained data? You should compare the dry weight of fungal biomass between the treatments.
Student´s t test was used to analyse significant differences between samples. A phrase was added to materials and methods section (lines 19-21)
In the case of dry weight biomass, no significant differences were found only in the first 6 h between 0.5 and 1% glycerol cultures.
- Line 195: on what basis was it found that the differences are insignificant?
We apologize for the expression, we wanted to say that the cultures followed a similar behaviour in both cases, with and without buffer. The phrase has been changed accordingly (line 209).
- Line 237: what is your hypothesis about the highest production of enzyme under carbon starvation conditions?
This discovery has been discussed before (Méndez-Líter et al., 2018), where BGL-3 production was studied in detail. The main hypothesis considered that induction of hydrolases, including glycosidases, is a key event in the aging of fungal cultures during carbon shortage. The high activity of BGL-3 on β-1,3 polysaccharides suggested its possible physiological role in cell wall metabolism during carbon starvation. That way, T. amestolkiae could use the fungal autolysis products as an alternative carbon source. This has also been suggested for other b-glucanases of Phanerochaete chrysosporium where b-glucosidase activity and the metabolism of the fungal cell wall was also proposed (Igarashi et al., 2003). In a near future, we would like to study T. amestolkiae aged cultures in detail for the discovery of novel and interesting catalysts with industrial applications.
A new sentence about the possible role of BGL-3 in aged cultures has been introduced lines 254-257.
Méndez-Líter JA, de Eugenio LI, Prieto, A., Martínez MJ. The β-glucosidase secreted by Talaromyces amestolkiae under carbon starvation: A versatile catalyst for biofuel production from plant and algal biomass. Biotechnol Biofuels. 2018;11:1–14.
Igarashi K, Tani T, Rie K, Masahiro S. Family 3 β‑glucosidase from cellulose‑degrading culture of the white‑rot fungus Phanerochaete chrysosporiumis a glucan 1,3‑β‑glucosidase. J Biosci Bioeng. 2003;95:572–6.
- line 249: why did you choose 1% of glycerol for proteomics? why only one condition?
We decided to analyse data from one condition in order to compare them with previous studies where we have used 1% of different carbon sources (de Eugenio et al., 2017). However, the suggestion of considering the proteomics analysis of different glycerol concentrations is very interesting, and we will take it into account for future works.
- What about the fact that activity of enzyme is vulnerable to pH changes? Is it a limitation in its use in waste management? Please consider this in the Conclusion section.
In fact, data obtained in this work suggest that pH changes in media strongly affect protein secretion, which translates into enzymatic detection, as described in Figures 2 and 3. pH should be controlled in order to obtain the highest protein/biomass ratios. Although pH influence in BGL-3 activity has not been reported in this article, previous works reported the high stability of this enzyme, retaining more than 80% residual activity after 72 h at pH 7 (Mendez et al. 2018).
Méndez-Líter JA, de Eugenio LI, Prieto, A., Martínez MJ. The β-glucosidase secreted by Talaromyces amestolkiae under carbon starvation: A versatile catalyst for biofuel production from plant and algal biomass. Biotechnol Biofuels. 2018;11:1–14.
Overall comment: you can improve the charts - maybe add some colors or more varied markers.
Charts aesthetics have been improved, adding colours.
Round 2
Reviewer 2 Report
Accept in the present. TIC does not make any sense. Data should be more clear.
Reviewer 3 Report
Dear Authors,
Thank you for your improvements. Now the manuscript sounds better and is easier for readers.